# The Impact of Person-Centered Care Indicators on Care Strain Among Care Aides in Long-Term Care Homes in New Brunswick: A Cross-Sectional Study

**DOI:** 10.3390/nursrep15050140

**Published:** 2025-04-26

**Authors:** Patricia Morris, Jennifer Moore, Rose McCloskey, Karen Furlong

**Affiliations:** 1Faculty of Nursing, University of New Brunswick, Fredericton, NB E3B 5A3, Canada; 2Long Term Care Simulation Lab, University of New Brunswick, Saint John, NB E2K 5E2, Canada; jmoore10@unb.ca; 3Department of Nursing and Health Sciences, University of New Brunswick, Saint John, NB E2K 5E2, Canada; rmcclosk@unb.ca (R.M.); kfurlong@unb.ca (K.F.)

**Keywords:** long-term care, dementia care, person-centered care, care strain, cross-sectional study, nursing assistants

## Abstract

**Background:** Person-centered care (PCC) approaches are widely recognized for improving the quality of life of residents living with dementia in long-term care (LTC). However, residents are only one part of the care dyad, and it remains unclear whether PCC also impacts nursing care aides in similarly adventitious ways. Care aides in this context experience significant care strain, which refers to the physical, emotional, and psychological burden experienced by caregivers. While PCC approaches are promoted as the best approach for supporting residents living with dementia, there is limited research on whether their implementation also impacts care aides. This study examined potential associations between organization-level PCC indicators and care strain among nursing care aides who work with residents living with dementia in LTC homes in New Brunswick, Canada. **Methods:** A cross-sectional survey design was used to explore the relationship between PCC approaches and care strain. Care strain was measured using the strain in dementia care scale, including the daily emotions subscale. A modified version of the Dementia Policy Questionnaire assessed the extent to which PCC approaches were implemented in participants’ workplaces. Descriptive statistics characterized the sample, and multivariable regression analyses examined associations between PCC indicators and care strain, adjusting for demographic factors. **Results:** Twenty-eight participants completed both measures. Overall, participants reported high levels of care strain but also high levels of positive daily emotions. Findings partially supported the hypothesis that PCC indicators were associated with lower care strain and more positive daily emotions. Certain PCC indicators, such as structured education and ethical support, appeared particularly beneficial. Implications for Practice: Strengthening PCC practices—especially through hands-on training and ethical support—may help reduce care strain and enhance care aides’ emotional well-being. LTC facilities that prioritize these strategies over policy implementation alone may improve both staff well-being and quality of care for residents.

## 1. Introduction

Nursing care aides (also called nursing assistants, personal support workers, resident attendants) provide the majority of the hands-on nursing care to residents. They are responsible for assisting residents with activities of daily living, such as bathing, dressing, feeding, and mobility support. Research indicates that nursing care aides experience significant care strain [1,2,3]. Care strain refers to the physical, emotional, and psychological burden experienced by caregivers [4]. It is the stress that caregivers feel as they manage the complexities of caregiving. One key driver of increasing care strain for nursing care aides in long-term care (LTC) homes is the rising prevalence of dementia among LTC residents [3]. Behaviors associated with dementia, like resistance to care, sexual disinhibition, wandering, and aggression, all contribute to significant care strain [5], as care aides must navigate unpredictable and often distressing behaviors while maintaining residents’ dignity and autonomy. In response to these challenges, LTC homes have increasingly promoted person-centered care (PCC) for people living with dementia as the best approach to ensuring dignity and autonomy [6,7]. PCC has been widely shown to enhance the quality of dementia care by focusing on residents’ unique needs, preferences, and life histories [8]. While PCC is widely regarded as beneficial for residents [5,8], there has been limited exploration of whether its implementation also reduces care strain for care aides. Understanding how PCC affects both residents and formal caregivers is crucial for determining whether this approach is sufficient for supporting each part of the caregiving dyad.

This study aimed to examine the relationship between PCC approaches and nursing care aide strain while working with residents living with dementia in LTC homes in New Brunswick. We were also interested in the relationship between PCC approaches and care aides’ self-reports of the daily emotions they experience at work. This study hypothesized that the presence of person-centered education, training, formal documentation practices, and supportive policies would each be associated with lower levels of care strain among care aides working with residents living with dementia. Additionally, we hypothesized that organizational expectations that care be provided, irrespective of available resources or residents’ dispositions, would be associated with higher levels of care strain.

## 2. Materials and Methods

### 2.1. Study Design

This study was a cross-sectional survey design. The STROBE checklist [9] for cross-sectional studies was consulted for comprehensive reporting and is available as a Appendix A. The surveys conducted for this study were administered as part of a larger mixed-method project that aimed to better understand how caring for residents living with dementia impacts the health and well-being of care aides. The larger study leveraged simulation as a research methodology, with participants engaging in a short simulation experiment with live actors who introduce strategically planned resistance to care to measure the physiological impact on care aides. The survey data were collected before the simulation began. Surveys were administered individually, in person, and completed using pen and paper. Surveys were completed in the conference room adjoining the simulation laboratory, where the rest of the study took place. Data were collected from June 2024–August 2024. An a priori G-power analysis (Anova, repeat measures, within factors) was performed to determine the ideal sample size for the larger study [10]. Twenty-four participants were required to achieve the desired power of 0.90 for the larger study. Post hoc power analysis was performed for each test conducted in this smaller study analysis, with an average power of 0.53.

### 2.2. Participants and Recruitment

Participants were recruited via a poster that was circulated for dissemination in local nursing homes. The poster was also shared via social media. The posters offered a 50$ VISA gift card as a token of gratitude for participants’ time. Snowball sampling and direct referrals were also employed after each person participated in the study. Eligible participants were 18 years of age and up, working full-time, part-time, or casually at a New Brunswick nursing home. Only participants who were able to come to the site where the simulation was completed were eligible for admission.

### 2.3. Ethics

Twenty-eight participants participated in the study. There was no attrition, though participants were told that they could stop the simulation and/or the surveys at any time. Upon arrival at the study site, participants were informed of the entire study protocol, and written informed consent was obtained. This study was reviewed by the University of New Brunswick Research Ethics Board (REB #2023-064) prior to commencement. The study protocol was described by a project manager for the study, who has no connections with the long-term care community in New Brunswick and would thus exert minimal influence over participant participation. This consideration was paramount in New Brunswick, which is a small province with fewer than 100 LTC homes. Following consent, participants completed questionnaires about their level of care strain, their emotional responses to working with residents living with dementia, and the PCC indicators present in their workplaces. Participants were given the choice to complete their surveys verbally with the project manager or to complete the surveys independently, using pen and paper, while the project manager waited in an adjacent room. All participants opted to complete the surveys independently. As self-report data carries a risk of social desirability bias, we aimed to minimize this by allowing participants to complete surveys privately. While the option for verbal survey administration was offered to reduce barriers to participation, we acknowledged the increased potential for social desirability bias in this mode. To mitigate this, the project manager was trained to deliver questions neutrally, and a physical shield was made available to allow participants to record their own responses privately, even when questions were read aloud.

### 2.4. Research Questions

The primary objective of this study was to examine the relationship between PCC approaches and nursing care aide strain while working with residents living with dementia in LTC homes in New Brunswick. The secondary objective was to examine the relationship between PCC approaches and care aides’ self-reports of the daily emotions they experience at work. The research questions that guided this project were as follows:In nursing care aides working with residents living with dementia in long-term care homes in New Brunswick, what is the relationship between organizational indicators of PCC approaches and levels of care strain?In nursing care aides working with residents living with dementia in long-term care homes in New Brunswick, what is the relationship between PCC approaches and the frequency and nature of self-reported daily emotions experienced at work?

### 2.5. Variables

Information was collected about demographics, strain in dementia care, daily emotions experienced at work, and the presence of PCC indicators in participants’ workplaces.

Demographics. Participants were asked to complete a short demographic questionnaire that focused on age, gender, education level, length of time in role, and work status (full-time, part-time, casual (as-needed)).

Strain in Dementia Care Scale. The strain in dementia care scale [11] (SDCS) is a validated 27-item tool with high internal consistency (Cronbach’s alpha = 0.96) and high test–retest reliability (0.88). An overall score of staff strain is calculated by averaging the product of two 4-point scales, one which assesses frequency of various dementia care situations or feelings (‘never’, ‘sometimes’, ‘quite often’, ‘very often’), and one which assesses the stress associated with each situation (‘none’, ‘mild stress’, ‘moderate stress’, ‘high stress’) [11]. Scores were also calculated for each of the 5 factors of strain that are represented by subsets of the items: (1) frustrated empathy (7 items), (2) difficulties understanding and interpreting (7 items), (3) balancing competing needs (5 items), (4) balancing emotional involvement (4 items), (5) lack of recognition (4 items). Scores range from 0 to 16, with higher scores indicating higher strain.

Daily Emotions. This scale is the second part of the Strain in Dementia Care Scale [12]. The scale requires that respondents report the frequency (on a 6-point scale ranging from “never’ to ‘all the time’) that they experience the following emotions during a day of work: powerlessness, satisfaction, sadness, frustration, fear, joy/happiness. An overall score was calculated by reverse-scoring the four negative emotions and summing the item scores for a total out of 30, with higher scores indicating more positive emotions (Cronbach’s alpha = 0.44). Since the daily emotions scale measures multiple, independent emotions rather than a single construct, a low Cronbach’s alpha is expected. The designers of the SDCS emphasized the importance of the two-fold response format to gather information on strain as well as the frequency of different emotional experiences [11].

PCC Indicators. This questionnaire was adapted from the Dementia Policy Questionnaire (Dem-Pol-Q) [13]. Respondents reported the presence of five indicators of PCC (four positive and one negative) in their workplaces by selecting ‘yes’, ‘no’, or ‘unsure’. These indicators, drawn from the Dem-Pol-Q, focus on structural and institutional factors that support the provision of PCC. The Dem-Pol-Q was designed to assess how organizations implement PCC approaches [13]. While it is a newer questionnaire, without an available Cronbach’s alpha, its construct validity was recently tested in two large studies [13,14]. While a number of studies have focused on how PCC is operationalized across care settings [15,16,17,18], the Dem-Pol-Q is currently the only questionnaire that measures PCC indicators in dementia care.

### 2.6. Data Pre-Processing

Data were extracted and uploaded into the statistical management software manually by the project manager and checked by two team members to ensure accuracy in extraction. The number of categories in the demographic variables was reduced to two or three categories per question for the bivariate analysis. This was performed by combining related categories (e.g., college education and university education were combined to make a post-secondary education category). This was performed to avoid categories with one or zero people occurring as a result of the low sample size for the study. Missing data were handled using removal. One survey contained missing data in two items from the SDCS scale. In this case, a total SDCS score was calculated without those two items. Three surveys were missing a majority of the data in the second column of the SDCS. Given that both columns must be multiplied together to obtain a total score, a total SDCS score could not be calculated for those three participants. The demographic, daily emotions, and PCC indicators questionnaire contained no missing data.

### 2.7. Statistical Analysis

Descriptive statistics were obtained for demographic variables and questionnaire responses using percentages for categorical variables and means and standard deviations for continuous variables. Inferential statistics were conducted in two phases using linear mixed models produced with the stats package in R v4.1.3 [19]. Linear regression was chosen as the analysis method for both the bivariate and multivariate models. Post hoc power analysis revealed relatively low power for the statistical tests. The average post hoc power of the tests run on this data was 0.53. Bayesian regression was considered as an alternative as it is more robust to small sample size, but there was insufficient prior knowledge to apply an a priori distribution that would increase the robustness over linear regression. Since the prior distribution that is chosen has a large influence on the results when the sample size is small, it was decided to use low-powered linear regression rather than risk a distorted result due to Bayesian regression with ill-informed priors [20]. While it is common for small sample sizes to cause model assumptions to be violated, necessitating the use of non-parametric alternatives, no assumptions of linear regression were violated by any of the models. There were no deviations from normality in either dependent variable, as assessed using the Shapiro–Wilk test (alpha = 0.05). There were no deviations from normality in the residuals of any model (assessed using skewness (<2), kurtosis (<5), and Shapiro–Wilk test (alpha = 0.05)), and no indication of any outliers (>3 SD from mean) in any model. There was also no heterogeneity of variance in any model (assessed using Levene’s test (alpha = 0.05) for the policy predictors).

In the first phase, bivariate analyses were used to determine which demographic variables were related to SDCS and daily emotion scores. In order to maintain power in the multivariate models, only demographic variables that exhibited a significant relationship with the respective dependent variable were commuted to the second phase. In the second phase, four multivariate linear models were run for each dependent variable containing the significant demographic variables and one of the categorical PCC indicator variables as predictors. To account for the repeated tests on the same data, Bonferroni alpha-level correction was used. A *p*-value of 0.0125 was taken to be significant in all multivariate analyses.

## 3. Results

### 3.1. Descriptive

The participants were 28 care aides from multiple long-term care homes in New Brunswick, Canada. The sample was predominantly female (n = 26; 93%), with only one participant self-identifying as male (n = 1; 3.5%) and one participant self-identifying as trans (n = 1; 3.5%). Ages ranged from 19 to over 65, with an even distribution across the age groups. Most had college diplomas in a related field (n = 18), but two participants had only a high school diploma, five had a university degree, and three had some other kind of education. The sample had worked in a long-term care role for an average of 7.3 years, with seventeen participants working full-time and the remainder working part-time.

The mean strain in dementia care score for the care aides in this study (n = 25) was 6.69 (±2.75) out of a possible 16, with higher scores showing more strain. While there is no agreed-upon cut-off for what constitutes a high score, this average score is higher than many other studies that use the scale (range 2.7–5.7) [21,22,23,24,25]. Of the five factors within the scale, care aides scored the highest on the frustrated empathy factor (8.07 ± 3.41), followed by lack of recognition (7.32 ± 3.98), balancing competing needs (7.28 ± 3.74), balancing emotional involvement (7.03 ± 3.25), and difficulty understanding and interpreting (4.38 ± 1.71). The questions that elicited the highest strain scores (>8) were the following:14. I have to prioritize on the basis of urgency rather than fairness or the needs of residents (8.16)15. I feel the residents are highly dependent on me. (8.13)19. The families of residents do not seem to understand how difficult it is to care for their relative. (8.68)22. I see that a resident is suffering. (8.57)

The second part of the SDCS revealed an average daily emotions score of 24.42 ± 6.06 (out of a possible 30), with higher scores indicating more positive emotional outlooks. When paired with descriptive results from the first part of the SDCS, our results suggested that participants maintain a relatively positive emotional outlook in spite of high strain in their workplace.

For PCC indicators, the majority of care aides responded ‘yes’ to having each of the indicators in place at their home, but an average of four care aides were unsure of the presence of each policy (Table 1). Item five was removed from later stages of the analysis because every respondent provided the same answer; all care aides reported that staff are expected to complete all personal care for their assigned residents.

### 3.2. Bivariate Analysis

Four demographic variables (age, education, full/part-time status, and years working as a care aide) were considered in the bivariate analysis using linear regression. None of the demographic variables showed a significant bivariate relationship with strain in dementia care. Part-time work status was associated with higher daily emotions (t = 2.538, *p* = 0.017). Work status was included as a covariate in the multivariate analyses due to the indication that it accounts for systematic variability in the dependent variables.

### 3.3. Multivariate Analysis

When controlling for work status, strain in dementia care was related to the availability of PCC training (f^2^ = 3.219, *p* = 0.003 and ethical guidance (f^2^ = 2.46, *p* = 0.004), but not person-centered documentation (f^2^ = 0.011, *p* = 0.889) or person-centered policies (f^2^ = 0.125, *p* = 0.329) (Figure 1). Similarly, daily emotions were related to the availability of PCC training (f^2^ = 1.169, *p* = 0.005) and ethical guidance (f^2^ = 2.039, *p* = 0.002), but not person-centered documentation (f^2^ = 0.018, *p* = 0.803) or person-centered policies (f^2^ = 0.085, *p* = 0.400) (Figure 2).

The presence of the measured PCC indicators exhibited the same pattern on the strain in dementia care as they did on participants’ daily emotions scores. The presence/absence/unknown of person-centered documentation and policies was not associated with deviations from the average strain (6.69 ± 2.75) or daily emotions (24.42 ± 6.06). However, in the case of PCC training and ethical guidance, care aides who were unsure about the availability of these indicators scored the most negatively on strain and emotion scales, while care aides who reported having these resources available scored the most positively. For example, care aides who reported access to guidance during ethically challenging situations had below-average strain scores (5.44 ± 1.79) and near maximum daily emotions (27.33 ± 4.65). The strain was slightly higher (6.42 ± 2.44), and emotions were slightly lower (23.00 ± 4.24) when care aides reported that ethical guidance was not available. But when care aids were not sure about whether they had access to ethical guidance, the strain score was exceptionally high (9.99 ± 2.42) and emotions were very low (16.83 ± 4.91). A nearly identical pattern was observed for the strain and emotions associated with the presence/absence/unknown of PCC training, in that those with access to PCC training had the most positive responses. The absence of PCC training was associated with higher strain and lower emotions but, when the availability of PCC training was unknown, strain was significantly higher, and emotions were significantly lower (Appendix A).

## 4. Discussion

Although the focus of this study was on care aides, findings have important and direct implications for nurses. As leaders in LTC [26,27], nurses are not only responsible for residents, but also for supporting, guiding, and overseeing care staff [27,28]. When care aides experience job strain and feel compelled to provide unwanted care to residents, it can lead to challenging work environments where both staff well-being and resident autonomy are compromised [29,30]. In these situations, nurses play a critical role in balancing both the needs of residents and the well-being of staff [31]. It is important for nurses to remain attentive to the emotional, physical, and mental impacts that care aides may experience as a result of caring for residents. Nurses should recognize that a well-supported and resilient workforce is critical to providing a high standard of care [29]. The higher-than-expected strain scores seen in this study underscore the need for nurses to prioritize the well-being of care aides alongside resident care. The finding that strain scores are high in part due to feelings of urgency in caring for residents and a lack of understanding among families about the challenges in caring for residents highlights specific areas where nursing interventions could help to alleviate care aides’ strain. For instance, nurses can create work environments that balance tasks with relationships, ensuring staff are not overwhelmed by unrealistic work demands. Additionally, nurses can work with families to help them develop clearer understandings of the complexities of resident care, setting realistic expectations for care aides.

While beyond the scope of this study, there is evidence that nurses are not always prepared for their supervisory role ([27], and do not always have clear guidelines or role descriptions [12,32]. Like care aides, nurses in LTC work under challenging conditions, carrying substantial responsibility for residents with complex medical needs while also overseeing large teams of unregulated care staff [28,33]. To better support nurses—and, in turn, care aides—system-wide reforms in the LTC sector are necessary to strengthen nurses’ ability to shape their work environment [34]. This includes actively involving nurses in organizational decision-making processes related to staffing, care protocols, and policy development [35,36]. According to Bowers et al. [34], strong nursing leadership is essential for improving workplace conditions, and empowering nurses in their supervisory roles can help address many of the challenges faced by frontline care staff. Future studies should examine how nurses’ supervisory abilities influence care aides’ PCC practices and work strain.

An unexpected finding from this study was that all participants noted an expectation to complete all personal care for assigned residents. While this factor was initially considered to be a PCC indicator, the overwhelming affirmative response to this item opens questions about refusal of care and whether or not care aides feel supported and equipped to truly respect a resident’s refusal of care. This finding is consistent with the literature, where care aides report significant barriers to respecting residents’ care refusals [37,38,39,40,41,42]. The care aides’ reluctance to honor the residents’ expressed care preferences highlights the important role nurses play in supporting and guiding care aides in the provision of PCC. Nurses’ clinical judgment and ethical decision-making skills are often required to navigate the often complex scenarios care aides find themselves in [43]. For example, there may be instances where nurses need to challenge existing rules and regulations in order to uphold PCC principles and support care staff.

It is important to note that approximately 15% of care aides in this sample were unsure whether PCC indicators existed in their workplace. Those who answered ‘unsure’ showed higher levels of care strain and poorer emotional outlook. This finding is consistent with the broader health literature that reports operationalization of PCC is most effective when staff are well-informed about how PCC applies in their specific practice area [16,44,45]. This finding further supports the importance of the nursing supervisory role in LTC as it applies to education, training, and compliance with institutional policies and philosophies [27,35,36].

Although this study was conducted within a single geographic setting with LTC staff, the findings may have broader relevance for similar care environments in other countries. The challenges faced by care aides and nurses—such as navigating complex care situations, managing emotional and physical strain, and making ethical decisions—are not unique to one context. International research has highlighted comparable issues across other settings [46,47,48], suggesting that systemic issues such as workplace strain are common globally [34,43]. As such, the experiences highlighted in this study may reflect a broader trend of workforce challenges across healthcare settings, emphasizing the need for cross-contextual strategies that not only support care providers and enhance care quality but also foster nurses’ leadership capacity.

### Limitations

The largest limitation of this study is the low sample size. Post hoc power analysis revealed relatively low power for the statistical tests that were run. The average post hoc power of the tests run on this data was 0.53. Nevertheless, there is reason to believe that there were no false negatives in these results because of the large difference in effect sizes between significant and non-significant results. The policies that acted as significant predictors had large effect sizes (Cohen’s f^2^ > 1.0). The policies that were not significant predictors of strain or emotions had effect sizes that were small enough (Cohen’s f^2^ < 0.1) that any reasonable increase in sample size would likely not have found a different result [49]. There were no policies that were on the edge of exhibiting a significant effect that were likely to have been classified incorrectly.

An additional limitation of this study is that the interpretation of the SDCS scores is variable. There is no agreed-upon cut-off for what constitutes a concerning score. Levels of strain were assessed only in relation to previous studies that have used the scale. Additionally, we did not directly assess care aides’ LTC homes for the presence of PCC indicators, so it is possible that they were available without the care aides’ knowledge or that the care aides overestimated their influence.

## 5. Conclusions

This study highlights the significant care strain experienced by care aides in LTC. Despite these high levels of strain, many care aides maintained a relatively positive emotional outlook, suggesting a dual experience of stress and fulfillment in their roles. The persistence of high strain scores despite the presence of PCC indicators highlights the ongoing systemic challenges in LTC environments. Future research should explore additional factors that influence caregiver well-being, including workload, staffing levels, and workplace culture, to better support care aides in providing high-quality, person-centered care.

Results of this study support the implementation of only two PCC indicators to reduce care aide strain in dementia care. First, mandated PCC training for all staff was related to a reduction in strain and improvement in daily emotions. This is consistent with the success of PCC training interventions on reducing care aide strain and stress [50,51]. Regardless of the type of training, mandating dementia care-specific training appears to be beneficial and in line with the requests from care aides for specialized care provided by their organizations [52]. A second PCC indicator identified as an important predictor of strain was having a process in place to provide guidance and support for staff dealing with ethically challenging resident care issues. Access to support at work has been shown to reduce burnout [53], making it a particularly important action.

## Figures and Tables

**Figure 1 nursrep-15-00140-f001:**
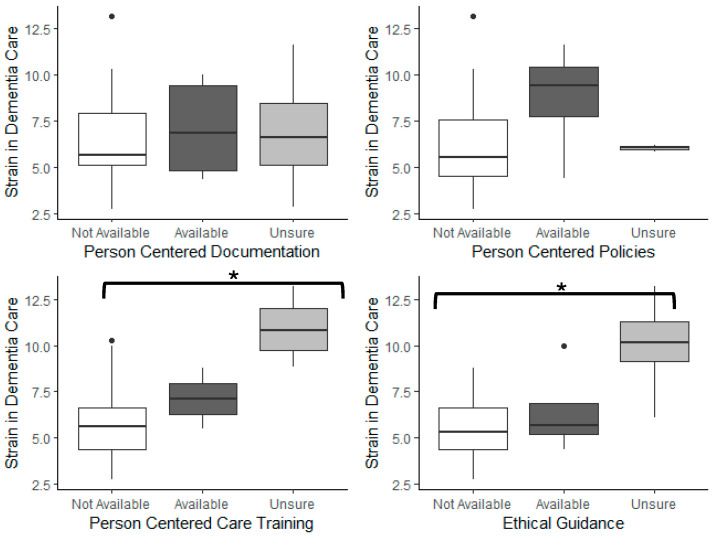
Relationship between strain in dementia care and PCC markers. Asterisks indicate significant differences (* *p* < 0.01).

**Figure 2 nursrep-15-00140-f002:**
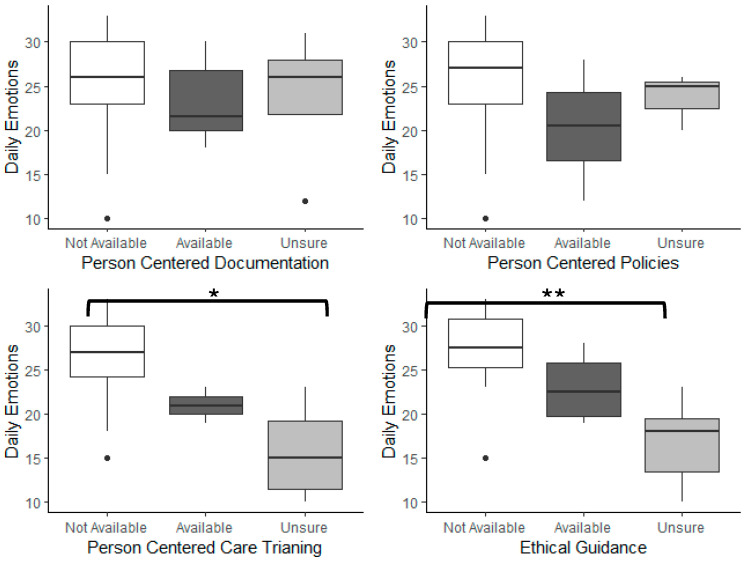
Relationship between daily emotions scores and PCC markers. Asterisks indicate significant differences (* *p* < 0.01, ** *p* < 0.001).

**Table 1 nursrep-15-00140-t001:** Presence of PCC indicators (n = 28).

Item	Yes	No	Unsure
1. Preferences of residents are recorded systematically and in a structured way.	18	6	4
2. There are policies/procedures in place on how to deal with refusals of care.	21	4	3
3. Mandatory education and training on person-centered care are required for all staff.	22	2	4
4. A process is in place to provide guidance and support for staff dealing with ethically challenging resident care issues.	18	4	6
5. It is expected that staff will complete all personal care (e.g., bathing) for the assigned residents.	28	0	0

Note: Items 1–4 align with PCC when they are present; item 5 aligns with PCC when it is absent.

## Data Availability

Interested parties are invited to contact the corresponding author for access to anonymized raw data used for this study.

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
