# Peer review of "The Impact of Person-Centered Care Indicators on Care Strain Among Care Aides in Long-Term Care Homes in New Brunswick: A Cross-Sectional Study"

_nursrep, 2025, doi:10.3390/nursrep15050140_

Round 1
Reviewer 1 Report
Comments and Suggestions for Authors
Dear Authors,
the comments in the annex file.
Best

Author Response
Thank you for your constructive feedback. Please find below a line by line response to your comments.
Comment 1: Title: Fine, but the term "markers" might be misleading. If I understood correctly, terms like "indicators" or "outcomes" may be more suitable. The same applies to "strain"; in this case, I think the term "stress" would be more accessible to the reader.
Response 1: We have replaced the term “markers” with “indicators” as per your suggestion. We also replaced the term “job strain” with the term “care strain,” since this term is used in the title of the scale used in this study. We define care strain in lines 16-17 and 50-51.
Comment 2: Abstract: I suggest opening the discussion on the possible implications for clinical practice (see discussions).
Response 2: We included a section titled “Implications for practice” in the abstract (lines 36-40).
Comment 3: Keywords: I suggest expanding to at least 4-5 terms and including the healthcare worker involved, as well as the type of study conducted.
Response 3: Done. New keywords are: long-term care; dementia care; care strain; cross-sectional study; nursing assistants
Comment 4: Introduction: Overall, it’s good, but there is redundancy (especially between lines 46-75); I suggest reducing it by at least 20% and focusing more specifically on the main points of the study.
Response 4: Done. The introduction is now approximately 20% shorter and focuses more specifically on the main points of the study.
Comment 5: Be careful with line 35, where there is an acronym not defined.
Response 5: A CTRL+FIND was performed on all acronyms used in the paper to ensure that they were defined at the beginning of the writing.
Comment 6: In the same sentence ending at line 36, I suggest clarifying the concept from a “legislative/administrative” perspective, as well as the required level of education and training, which might be specific to one context and not homogeneous in others. I believe it refers to support staff for nursing personnel.
Response 6: Removed this section and focused more specifically on the main points of the study.
Comment 7: Objectives: Poorly defined and unclear. I suggest using the traditional structure of explicitly stating the primary objectives and then the secondary ones, in the usual formula “the primary objectives of the study were... while the secondary ones...”. I would also add the research questions that the researchers aim to answer, and perhaps dedicate a specific section in the methods, e.g., 2.1 “Aims and research questions”. Response 7: Objectives have been restated in the traditional structure you have proposed (lines 65-69). A section was added to the Methods section titled “Research questions” where research questions are stated explicitly (lines 123-131).
Comment 8: Methods: This is the most controversial aspect and certainly requires more attention, particularly the lack of a structured reporting method, such as the STROBE checklist (doi:10.1016/j.jclinepi.2007.11.008), which is widely considered mandatory by the scientific community. Including this checklist as a supplementary file and citing it in the text would be essential for improving the quality and reproducibility of the study, making the manuscript more transparent and scientifically valid.
Response 8: Referenced the STROBE checklist; followed the reporting guidelines to ensure each point was discussed (lines 83-84).
Comment 9: Results: Good section, it will certainly benefit from the previous and subsequent suggestions. It is certainly the best-built part, but it only requires attention regarding the proposed figures, which I suggest editing more clearly.
Response 9: Edited the figures for clarity.
Comment 10: Discussion: This is the section that requires the most attention, as it is not supported by relevant references and is repetitive in terms of results in the first part (lines 258-281). I suggest reducing it to a simple summary from which to begin the discussions and comparison with the literature. I understand the difficulties in addressing the discussion concerning this support role, but given the journal’s topic, I believe it is necessary to extend the discussion to nursing staff, perhaps in different contexts in terms of settings or countries, or similar to the setting of the study. In this regard, I would like to suggest some highly relevant articles: doi: 10.7417/CT.2020.2233 (for different settings and countries), doi: 10.1097/HNP.0000000000000512 (similar care setting but in a different country), doi: 10.1017/S0714980818000478 (similar setting and country, and also useful as a reference institutional profile for the researchers).
Response 10: The discussion has been modified significantly to address these concerns. The summary has been shortened to focus on the hypotheses for the study (line 297-301), followed by a discussion of unexpected findings and connections with the broader literature.
Comment 11: Additionally, I suggest creating a specific section. The last section, which I would honestly call “4.2 Perspectives for Clinical Practice,” could benefit from expanding the discussion on the proper multi-dimensional and multidisciplinary management of patients in long-term care.
Response 11: This section has been added (4.1), lines 333-344.
Comment 12: Limitations and Conclusions: I suggest creating a specific section for limitations (4.1 Limits), perhaps before the previous section, and separating it from the conclusions.
Response 12: A limits section is now included (4.2) (lines 345-362).
Comment 13: Bibliography: It should be expanded according to the previous suggestions, and it might be helpful to update references older than 15-20 years, unless they are methodological or have strong impact evidence.
Response 13: All references have been reviewed for their relevance to the paper. Texts that are more than 15 years old were kept only if they are considered seminal works in this field of study (e.g. Kitwood’s text on person-centered care) or if they were examples of studies where the SDCS was used (e.g. Orrung et al., 2013)
Thank you for your feedback.
Reviewer 2 Report
Comments and Suggestions for Authors
The study gives relevant information about job strain among care aides in long-term care homes. The number of references is good. The gap in knowledge is clearly identified using references. The references are mostly quite new, and the references are relevant. However, many of the references are quite old. Many newer references of the field exist as job well-being in healthcare is very important research topic. The scales of the study are presented clearly and are well justified. The results are reported quite clearly using figures. The amount of respondents is very low, which is a huge limitation of the study even the authors have stated. The low number of respondents limit the possibilities of statistical analyses. Descriptive analytics, e.g. frequencies, should present in amounts, not percentages. Some other methods instead of factor analysis would have been better for such a small sample size. In the discussion the authors present the findings and compare the results with previous studies very well. The statements and conclusions are coherent, and they are supported by the listed citations.
Line 182: Please don't use percentages for 28 respondents.
Line 218: What kind of bivariate test was used?
Line 250: The texts in figure 1 are a bit unclear.
Line 253: Figure is a bit unclear, the asteriks and text are difficult to read.
Author Response
Thank you for the constructive feedback on this report. We have included responses with line numbers indicated in response to each comment.
Comment 1: The study gives relevant information about job strain among care aides in long-term care homes. The number of references is good. The gap in knowledge is clearly identified using references. The references are mostly quite new, and the references are relevant. However, many of the references are quite old. Many newer references of the field exist as job well-being in healthcare is very important research topic.
Response 1: All references have been reviewed for their relevance to the paper and newer literature was introduced where appropriate. Texts that were more than 15 years old were kept only if they are considered seminal works in this field of study (e.g. Kitwood’s text on person-centered care) or if they were examples of studies where the SDCS was used (e.g. Orrung et al., 2013).
Comment 2: The scales of the study are presented clearly and are well justified. The results are reported quite clearly using figures. Descriptive analytics, e.g. frequencies, should present in amounts, not percentages.
Response 2:This has been modified. See Table 2 (p. 5-6).
Comment 3: The amount of respondents is very low, which is a huge limitation of the study even the authors have stated. The low number of respondents limit the possibilities of statistical analyses. Some other methods instead of factor analysis would have been better for such a small sample size.
Response 3: Factor analysis was not used in this study. Linear regression was chosen as the analysis method for both the bivariate and multivariate models. We recognize that it is common for small sample sizes to cause model assumptions to be violated, necessitating the use of non-parametric alternatives. However, no assumptions of linear regression were violated by any of the models. Our reasoning is outlined in narrative format in the paper, on lines 194-210.
Comment 4: In the discussion the authors present the findings and compare the results with previous studies very well. The statements and conclusions are coherent, and they are supported by the listed citations.
Response 4: Thank you.
Comment 5: Line 182: Please don’t use percentages for 28 respondents.
Response 5: Amended.
Comment 6: Line 218: What kind of bivariate test was used?
Response 6: Linear regression. Now specified in the text (line 194)
Comment 7: Line 250: The texts in figure 1 are a bit unclear. Line 253: Figure is a bit unclear, the bstract and text are difficult to read.
Response 7: Amended.
Thank you for your thoughtful contributions to this report.
Reviewer 3 Report
Comments and Suggestions for Authors
Thank you for allowing me to review your work. I have attached some comments. I would suggest clearly defining your variables and how they align with your instruments so that it is clear when you are discussing, and making sure you introduced the rationale for these variables.

Author Response
Thank you for your feedback on this report. Please see below a comment by comment response including line numbers for reference.
Comment 1: Abstract: Were the variables discussed on line 11-12 measured at all in the surveys? How are they relevant to the study in terms of your outcomes? You mention time pressures on line 59 but do not clarify the extent of the significance. If there is a significance for these factors, I think a little more information is needed in introduction section to build up why this is significant
Response 1: The abstract has been refined to focus specifically on the variables measured in the surveys (care strain and PCC indicators).
Comment 2: Keywords: your keywords don’t quite reflect what you are doing. I think you should make your key words more in alignment with the aim
Response 2: New keywords are: long-term care; dementia care; person-centered care; care strain; cross-sectional study; nursing assistants
Comment 3: Introduction: Please clarify what is meant by unregulated on line 35.
Response 3: The introduction has been amended to focus more specifically on the role of the care aide. An unregulated care aide refers to a staff member who provides direct care but does not have a governing regulatory body overseeing their practice. Despite this, their work is often structured by strict routines and staffing ratios set by the organization to ensure efficiency and consistency in care delivery.
Comment 4: Please clarify what BPSD are. You do a nice job indicating the prevalence and impact but not quite in describing what these symptoms are and how they are managed, how they disrupt the workload of the care aides. Is rushed care a consequence of BPSD?Response 4: This section of the introduction has been significantly amended based on another reviewer’s feedback to refocus the introduction on the main variables of interest in this study. As a result, discussion of BPSD was decreased in favour of a broader focus on dementia care and challenging behaviours associated with dementia (lines 54-55).
Comment 5: Your aim (line 88-92) is not in alignment with the aim in the abstract. You are discussing how organizational factors mitigate the stress on care workers and the buildup seems to be specific to the reduction of strain from PCC. The abstract indicates an exploration of factors that impact strain, suggesting multiple factors. I suggest congruence.
Response 5: The abstract has been amended to reflect the aim of the study.
Comment 6: Methods: Please clarify what casual employment is for the inclusion criteria.
Response 6: Line 141 (included synonym “as-needed”)
Comment 7: Identify how many participants were accessed, how many completed the study. How were surveys administered? Was there any attrition? Were participants able to stop the simulation and/or the survey at any time?
Response 7: Now included. Lines 108-110 and 116-122.
Comment 8: Scales: Please identify why the instruments were chosen. Are these the two instructions that represent those variables best in the literature? Were there other scales? Consistency is mentioned for 1 scale. Please add the reliability for all instructments as ewel as indicators for validity. Was the Dem-Pol-Q from the literature or from another source? Please add reliability and validity for this assessment. For Table 1, is there any other statistical information that can be provided with this table?
Response 8: Reason for choosing SDCS is described in lines 142-144, along with internal consistency and test-restest reliability. Daily emotions subscale is described similary in lines 156-162. Construct validity of Dem-Pol-Q and why it was selected is discussed in lines 165-172.
Comment 9: Data pre-processing: Was there a method for data extraction and reduction methods? Please add and cite if so.
Response 9: Included. Lines 177-178.
Comment 10: Statistical analysis: Please add the criteria for the p level and other statistical tests. Was power calculated beforehand? Was it achieved? If not, please list this as a limitation of the limitation of the study. If you performed F tests I am concerned that you may not have had enough participants for sufficient power
Response 10: P levels have been added in the table submitted as supplemental data. The statistical power issue is now addressed in the limitations section (4.2), where we discuss the post-hoc power analysis and reasons to believe that there were no false negatives in these results. We also discuss our reasoning for choosing linear regression in lines 195-211.
Comment 11: Figure 1: There is some kind of printing error in this that overlays your labels.
Response 11: Fixed.
Comment 12: Discussion: Are you looking to see how much strain there is for your participants? I am concerned about the introduction of the term frustrated empathy here in the discussion section that you did not speak about before. What was this representing? You brliefly idenfify the factor but not what it represents. I think if you have components that the scales are representing, it is important that you clearly identify and defined all of your operational variables before the discussion section so that the results and discussion are clear.
Response 12: Thank you for this direction. The discussion section has been amended significantly to focus on the variables of interest for this study, and all discussion of individual factors in the SDCS have been removed. We also took this opportunity to reiterate the hypotheses for the study and to elaborate on which were supported/unsupported.
Round 2
Reviewer 1 Report
Comments and Suggestions for Authors
Dear Authors,
the comments in the annex file.
Best

Author Response
Thank you for your continued feedback and attention to detail in making this a strong contribution to the scientific community.
Comment 1: The Strobe checklist, considered for reporting, is missing among the supplementary files, which is mandatory for the reference scientific community.
Our apologies. The STROBE checklist should have been submitted along with the other supplementary document. We have consolidated it into one document for this submission.
Comment 2: The comment on the objectives was partially perceived because the research questions were posed, but the objectives were not clearly stated.
See lines 121-125
Comment 3: In line 230, the bibliographic references are unclear, as they do not follow the correct order and are not listed according to the journal’s template.
Amended.
Comment 4: The comment on suggestion 10 in the discussion is also partial, as the authors only considered the "same setting and country" option, leaving the comparison with the other two elements indicated rather limited. I suggest considering these other elements to better broaden the discussion and elevate it scientifically in a clear and substantial way.
Thank you for your continued feedback. The discussion has been amended again to consider the setting and country options, and the discussion now speaks to broader themes that we hope sufficiently elevate it.
Reviewer 2 Report
Comments and Suggestions for Authors
You have revised your manuscript according to my comments and suggestions. You have made good corrections to your manuscript.
Author Response
Thank you.
Reviewer 3 Report
Comments and Suggestions for Authors
Thank you for your revisions. I have attached very few suggestions.

Author Response
Comment 1: Add a citation for the power analysis tool (G-power or another method).
Added.
Faul, F., Erdfelder, E., Buchner, A., & Lang, A.-G. (2009).
Statistical power analyses using G*Power 3.1: Tests for correlation and regression analyses.
Behavior Research Methods, 41(4), 1149–1160.
https://doi.org/10.3758/BRM.41.4.1149
Comment 2: Add any power imbalances related to verbal survey implementation to the methods
Added in lines 112-118.
Comment 3: Line 197, I appreciate the addition of statistical decisionmaking to support the results. Please add a citation for the justification of the statistical decision and any cut scores/critieria
Added.
Smid, S. C., McNeish, D., Miočević, M., & van de Schoot, R. (2020). Bayesian versus frequentist estimation for structural equation models in small sample contexts: A systematic review. Structural Equation Modeling: A Multidisciplinary Journal, 27(1), 131–161. https://doi.org/10.1080/10705511.2019.1577140
Comment 4: Table 2, Line 254, If any of these are reverse coded, please include a footnote
Footnote added, line 256.
Comment 5: Limitations, Line 349. Please add the Cohen reference to support effect size decisions
Added.
Cohen, J. (1988).
Statistical power analysis for the behavioral sciences (2nd ed.).
Hillsdale, NJ: Lawrence Erlbaum Associates.
Comment 6: Please add the statistical criteria for this study to the footnotes of the supplemental tables
Added.